# Collective intelligence defines biological functions in Wikipedia as communities in the hidden protein connection network

Andrei Zinovyev[1,2,3‡]*, Urszula Czerwinska[1,2,3], Laura Cantini[1,2,3,4], Emmanuel Barillot[1,2,3], Klaus M. Frahm[5], Dima L. Shepelyansky[5‡]

**1** Institut Curie, PSL Research University, F-75005 Paris, France, **2** INSERM, U900, F-75005 Paris, France, **3** MINES ParisTech, PSL Research University, CBIO-Centre for Computational Biology, F-75006 Paris, France, **4** Computational Systems Biology Team, Institut de Biologie de l'Ecole Normale Supérieure, CNRS UMR8197, INSERM U1024, Ecole Normale Supérieure, PSL Research University, F-75005 Paris, France, **5** Laboratoire de Physique Théorique, IRSAMC, Université de Toulouse, CNRS, UPS, F-31062 Toulouse, France

‡ These authors are joint senior authors on this work.
* andrei.zinovyev@curie.fr

**Data Availability Statement:** All relevant data are within the manuscript and its Supporting Information files.

## Abstract

English Wikipedia, containing more than five millions articles, has approximately eleven thousands web pages devoted to proteins or genes most of which were generated by the Gene Wiki project. These pages contain information about interactions between proteins and their functional relationships. At the same time, they are interconnected with other Wikipedia pages describing biological functions, diseases, drugs and other topics curated by independent, not coordinated collective efforts. Therefore, Wikipedia contains a directed network of protein functional relations or physical interactions embedded into the global network of the encyclopedia terms, which defines hidden (indirect) functional proximity between proteins. We applied the recently developed reduced Google Matrix (REGOMAX) algorithm in order to extract the network of hidden functional connections between proteins in Wikipedia. In this network we discovered tight communities which reflect areas of interest in molecular biology or medicine and can be considered as definitions of biological functions shaped by collective intelligence. Moreover, by comparing two snapshots of Wikipedia graph (from years 2013 and 2017), we studied the evolution of the network of direct and hidden protein connections. We concluded that the hidden connections are more dynamic compared to the direct ones and that the size of the hidden interaction communities grows with time. We recapitulate the results of Wikipedia protein community analysis and annotation in the form of an interactive online map, which can serve as a portal to the Gene Wiki project.

## Author summary

The long-standing effort for annotating protein functions from published experimental evidences is still far from being completed, partly due to a limited number of biocurators involved in it. Wikipedia was thought to be a suitable platform for the protein function

**Funding:** This research is supported in part by the MASTODONS-2016/2017 CNRS project APLIGOOGLE, in part (for KMF and DLS) by the Programme Investissements d'Avenir ANR-11-IDEX-0002-02, reference ANR-10-LABX-0037-NEXT (project THETRACOM) and by the French government under management of Agence Nationale de la Recherche as part of the "Investissements d'avenir" program, reference ANR-19-P3IA-0001 (PRAIRIE 3IA Institute). The funders had no role in study design, data collection and analysis, decision to publish, or preparation of the manuscript.

**Competing interests:** The authors have declared that no competing interests exist.

curation crowdsourcing through exploiting the wisdom of the crowd principle. Starting from 2008, English Wikipedia was automatically populated with thousands of protein pages and links between them (Gene Wiki project), which created a useful and rapidly evolving knowledge resource. However, it remains unclear what is the benefit of hyperlinking protein pages with the whole Wikipedia knowledge corpus. We applied the recently introduced network analysis method, called reduced Google Matrix (REGOMAX), in order to study the structure of direct and indirect (hidden) links between protein pages through the rest of the global Wikipedia network. As expected, the network of direct links had node degree distribution approximately following the power law. In contrast, the network of hidden links was characterized by larger than expected tight communities of proteins related to their known functions, such as involvement in immune system. The "friendship network" of these protein groups can be used for automated annotations of their functions from non-protein Wikipedia pages. We estimated the size of the expert Wikipedia contributor community, specifically working on protein and associated pages, to be nearly 1000 wikipedians with primarily biomedical background. We conclude that the structure of global Wikipedia network can improve the annotation of protein functions by amplifying the wisdom of the crowd effect.

## Introduction

Wikipedia is a unique knowledge resource containing a collection of approximately 5.5 millions articles in its English version, connected with each other by approximately 122 millions links (data from year 2017). Studying the large graph of Wikipedia hyperlinks with a focus on a particular subset of pages can provide interesting insights about certain topics. Thus, for example, Wikipedia networks were explored to establish the top historical figures of human history over 15 centuries [1], the geopolitical relations between countries [2], the leading world universities [3], world influence of infectious and cancer diseases [4, 5]. Hierarchical structure of Wikipedia was revealed through application of network community detection algorithms [6]. The variety of applications of Wikipedia in academic research was reviewed in [7, 8].

Wikipedia is a resource curated by a decentralized community effort, which also includes semi-automated page generation from other structured resources. With time, automatically generated pages are modified by the community and hyperlinked with the rest of the encyclopedia and external Internet. In this way, potentially any structured resource can be imported into Wikipedia, profit from continuous collective annotation by the Wikipedia contributors and eventually become tightly embedded into the global Wikipedia knowledge graph.

Such an effort was made in the past for representing human genes in Wikipedia, the Gene Wiki project, https://en.wikipedia.org/wiki/Gene_Wiki. By 2008, a massive import of approximately 8000 gene-specific pages from Entrez Gene database was made, which boosted the community-based annotation of genes [9]. The initial set of protein page "stubs" have been complemented by adding the knowledge about protein-protein interactions, represented by hyperlinks between pages. Thus, in 2009, 3389 protein pages were connected by the 12628 most confident interactions from BioGrid database [10]. In 2011, it was estimated that 10369 protein pages in Wikipedia were annotated by 37578 PubMed citations with about 200 new citations added each month [11]. In 2009, Wikipedia protein pages of Wikipedia have been edited with a rate of approximately 1000 non-bot edits/month [10]. In 2016, the Gene Wiki project was complemented by the mechanism of Wikidata for better structuring the infoboxes

of protein and gene pages [12]. As of today, Wikipedia pages related to proteins have become tightly integrated with the pages of common interest, describing diseases, drugs, biological functions, general culture phenomena. For example, the "BRCA1" page in Wikipedia is linked by such pages as "Oncogene", "Mastectomy", "Joe DiMaggio", "Breast cancer", "Carleton College", "DNA repair" (selected from the top 20 at https://en.wikipedia.org/w/index.php?title= Special:WhatLinksHere/BRCA1).

The Gene Wiki experiment on embedding a set of protein pages into the structure of WikiPedia representing both general and specialized knowledge, is of great interest in several aspects. Firstly, it is interesting to verify if it can initiate a collaborative effort for summarizing knowledge about protein function, which remains a tedious, slow and prone to various types of biases process. Wikipedia potentially provides a platform for application of the "wisdom of the crowd" (or, "collective intelligence") mechanism known to be effective in other fields, including network biology [13]. In applications of crowdsourcing to solving complex problems, it can be sometimes advantageous to combine efforts of experts of different levels [14]. It remains unknown if such an effect impacts the quality of Wikipedia curation, in particular, in the Gene Wiki project. Secondly, it is interesting to hypothesize that the definition of a protein function can self-emerge from the tighter integration of protein pages with the rest of Wikipedia. The structure of the global Wikipedia hyperlinks induces a metrics (similarity measure) between protein pages: therefore, groups of proteins can converge (or, diverge) with time to each other in the metric semantic Wikipedia space, as a result of the changes in the rest of Wikipedia. Therefore, this opens a possibility for a much larger set of Wikipedia editors to contribute to defining the protein functions in an indirect way; thus amplifying the wisdom of the crowd effect. The major question is to prove if such a phenomenon exists.

In this article, we study in details how the knowledge about interactions between proteins is represented in Wikipedia from the network structure and dynamics point of view. We focused on quantifying how this knowledge is interconnected with the rest of the encyclopedia, serving a constantly updated corpus of annotation texts. Since the major bulk of direct protein-protein interactions has been automatically imported from existing databases of molecular interactions, the principal interest is in studying the topology of hidden, indirect connections between proteins through the rest of the Wikipedia graph. In order to study this topology, we used the recently developed methodology of reduced Google Matrix, which was already applied before for inferring hidden causal relations in a subnetwork of interacting proteins, embedded into a global network of protein-protein regulations [15]. As of today, the reduced Google Matrix method is the only one able to precisely quantify indirect oriented connections in very large networks (with millions of nodes and hundreds of millions of edges).

The performed network analysis used the PageRank algorithm, which is at the foundation of the Google search engine [16, 17], and other properties of the Google matrix employed for analysis of various types of directed networks [18]. The recent approach of reduced Google matrix (REGOMAX) [19, 20] allows establishing indirect interactions between the selected nodes of interest taking into account all paths between these nodes via the remaining part of global network with a large number of nodes. This REGOMAX algorithm originates from the scattering theory of nuclear and mesoscopic physics and field of quantum chaos [19].

Using the REGOMAX formalism, we characterized the topology of hidden connections between proteins in Wikipedia and identified the features distinguishing this topology from the network of direct connections. Following the recipe of the well-known proverb "*Tell me who your friends are and I will tell you who you are*" we characterize the function of specific hidden protein communities using their "*friendship network*" of links.

Wikipedia, or similar resources, are sometimes criticized by stating that they represent the wisdom of a few rather than the wisdom of the crowd [21]. In order to highlight this aspect, we

analyzed in details the set of WikiPedia users contributing to the protein pages and the friendship network in order to confirm that its structure is the result of "the collective intelligence" or "the wisdom of the crowd". We also characterized the nature of expertise level distribution among these contributors.

## Materials and methods

### Direct network of protein connections

Global network of links between English Wikipedia pages was extracted using in-house web crawler, for 2013 and 2017 years. This procedure creates a link from Wikipedia article A to article B when there is at least one citation of article B in the text of article A (see details in [1]). Protein and gene pages have been identified by querying for the presence of "Infobox protein" and "Infobox gene" Wikipedia templates in the page text. Those pages not having any outgoing or incoming links have been filtered out.

### Google matrix construction

The Google matrix $G$ of a directed network with $N$ nodes (titles) and $N_l$ hyperlinks is constructed from the adjacency matrix $A_{ij}$ with elements 1 if node (title) $j$ points to title (node) $i$ and zero otherwise. The matrix elements have the standard form $G_{ij} = \alpha S_{ij} + (1 - \alpha)/N$ [16–18] where $S$ is the matrix of Markov transitions with elements $S_{ij} = A_{ij}/k_{out}(j)$ and $k_{out}(j) = \sum_{i=1}^{N} A_{ij} \neq 0$ being the out-degree of node $j$ (number of outgoing links); $S_{ij} = 1/N$ if $j$ has no outgoing links (dangling node). The parameter $0 < \alpha < 1$ is the damping factor. We use the standard value $\alpha = 0.85$ [17] noting that for the range $0.5 \leq \alpha \leq 0.95$ the results are not sensitive to $\alpha$ [17, 18]. For a random surfer, moving from one title to another, the probability to jump to any title is $(1 - \alpha)$.

The right PageRank eigenvector of $G$ is the solution of the equation $GP = \lambda P$ for the unit eigenvalue $\lambda = 1$. The PageRank $P(j)$ values give positive probabilities to find a random surfer on a node $j$ ($\sum_j P(j) = 1$). We order all nodes by decreasing probability $P$ numbered by PageRank index $K = 1, 2, \ldots N$ with a maximal probability at $K = 1$ and minimal at $K = N$. The numerical computation of $P(j)$ is done efficiently with the PageRank algorithm described in [16, 17].

### Reduced Google matrix algorithm

The REGOMAX algorithm is described in detail in [19, 20]. It allows one to compute efficiently a "reduced Google matrix" of size $N_r \times N_r$ that captures the full transitions of direct and indirect pathways happening in the full Google matrix between $N_r$ nodes of interest.

For the selected $N_r$ nodes their PageRank probabilities remain the same as for the global network with $N$ nodes, up to a constant multiplicative factor taking into account that the sum of PageRank probabilities over $N_r$ nodes is unity. The computation of $G_R$ provides a decomposition of $G_R$ into matrix components that clearly distinguish direct from indirect interactions: $G_R = G_{rr} + G_{pr} + G_{qr}$ [20]. Here $G_{rr}$ is given by the direct links between selected $N_r$ nodes in the global $G$ matrix with $N$ nodes. In fact, $G_{pr}$ is rather close to the matrix in which each column is given by the PageRank vector $P_r$, ensuring that PageRank probabilities of $G_R$ are the same as for $G$ (up to a constant multiplier). Hence $G_{pr}$ does not provide much information about direct and indirect links between selected nodes. The most nontrivial and interesting role is played by $G_{qr}$, which takes into account all indirect links between selected nodes appearing due to multiple pathways via the global network nodes $N$. The exact formulas for all three components of $G_R$ are given in [19, 20].

The efficiency of the REGOMAX approach has been demonstrated for various Wikipedia networks [2–5, 20], protein networks from SIGNOR database [15], and the multiproduct world trade network from UN COMTRADE database [3].

All matrix data and PageRank vectors for the reduced Google matrix of $N_r$ = 4899 proteins are available at [22] for Wikipedia versions of 2013 and 2017, together with the global Wikipedian networks.

### Network of hidden protein connections

The network of hidden protein connections is obtained from the component $G_{qr}$ of the reduced Google matrix $G_R$ by keeping only matrix elements being larger than a certain critical cutoff value. This value if determined from the condition of having the same connectivity value (number of nodes divided by the number of edges) in the largest connected components of both networks of direct and hidden protein connections.

### Defining hidden communities by clustering

For the networks of hidden protein connections we applied Markov Clustering Algorithm (MCL) implemented in ClusterMaker plugin for Cytoscape [23] with default parameters (Inflation = 2.0, Expansion = 2.0). Parameter analysis of the MCL algorithm application to the graph of hidden protein connections has been performed (S1 Fig). We showed that the clusters in the hidden protein connection network can be easily matched for different reasonable combinations of MCL parameters. This means that most of the conclusions reported in this study do not qualitatively depend on the exact choice of the clustering parameters (even though the number of detected communities and their sizes can vary). We observed that increasing Expansion parameter is usually detrimental for the biological significance of the clusters, while increasing Inflation improves the significance. However, increasing values of Inflation tends also to produce smaller clusters. From our analysis we concluded that the default combination of MCL parameters represents a good balance between biological significance and the cluster size.

### Functional enrichment analysis

The functional enrichment analysis was performed using ToppGene [24] and recapitulating the results in the form of an interactive web-page, available at [22]. In the automatically produced summary of the enrichment results for each hidden protein community, one of the reference sets per category is displayed but only if it overlapped with the query set in at least $k$ = 5 genes and only if the corrected for multiple testing q-value did not exceed $s = 10^{-8}$.

## Results

### Wikipedia networks of direct and hidden connections between proteins

We first listed all English Wikipedia page titles containing a description of a protein or a gene and having a link with at least one other Wikipedia page. This resulted in 4899 protein page titles from the global Wikipedia network of $N$ = 5416537 titles with $N_l$ = 122232932 hyperlinks (for 2017 version and $N$ = 4212493, $N_l$ = 101611732 for 2013 version [25]). We used these two snapshots because year 2017 signifies approximately a decade of existence of the Gene Wiki project, and year 2013 signifies the beginning of an active curation effort on the protein pages after the initial automated bot-based page and hyperlink insertion phase.

Using the global Wikipedia graph, we extracted the subnetwork of oriented direct hyperlinks between protein pages, which we will call in further the "network of direct links" between

proteins. Then, we applied the reduced Google Matrix algorithm in order to quantify hidden links between all pairs of proteins (see "Reduced Google matrix algorithm"). As a result, for each oriented protein pair, a weight was assigned representing the strength of the hidden connection through the rest of the Wikipedia network. We filtered out the hidden connections having small weights, by setting a threshold such that the resulting network would have the largest connected component (LCC) with the same average connectivity (number of nodes divided by the number of edges) as the LCC of the network of direct links. We will refer to the remaining links as "strong hidden connections" (or simply hidden protein links for short). The resulting number of edges is indicated in Fig 1B. Overall, strong hidden links tend to connect less proteins than the direct ones. Most of both direct and hidden links form one largest connected component (LCC) comprising more than 96-97% of total number of links. The number of direct and hidden links grew from 2013 to 2017 (in 4 years) of 14% and 32%, respectively, showing that the strong hidden connections form LCC increasing in size with time.

A noticeable structural difference between direct and hidden network concerns the number of their bidirectional links (when two protein pages point to each other reciprocally): 35% in the direct network and only 10% in the hidden one, see Fig 1. This might reflect the way the information about physical interactions between proteins was populated in Wikipedia, where the large part of interactions were considered non-oriented, so if protein A has protein B in the list of protein with which it interacts, then B should have A in its corresponding list.

Hidden protein links are explained by the existence of connected Wikipedia page sequences (paths) of hyperlinks through the rest of the Wikipedia network that connect two protein pages. We were interested in quantifying how many Wikipedia pages separated two protein pages associated via a strong hidden link. We found out that for the absolute majority of strong hidden links the shortest path length was equal to 2, compared to 3 or 4 for a randomly chosen protein pair, or to 5 and more for a randomly picked pair of Wikipedia pages (10000 page pairs have been sampled in order to estimate the distribution), see Fig 1C. The shortest path length itself, however, is not a good measure of hidden link between two nodes of the graph, since it does not reflect the global topology of the Wikipedia network, e.g., the total number of shortest paths, connecting two nodes. By contrast, the weight of the link, estimated through the application of the reduced Google matrix, reflects the global topology of the rest of network and the probability to arrive from one protein page to an other one via a random walk with restart through the rest of the graph.

## Comparing the networks of direct and hidden protein links with existing pathway databases

We checked how many links extracted from Wikipedia matches known regulations or physical interactions between proteins, described in existing pathway databases. With this aim, we compared the protein connection networks reconstructed from Wikipedia with two publicly available protein-protein interaction networks. One such database, SIGNOR, is characterized by a relatively small size and contains a set of highly confident interactions [26]. Another database, Pathway Commons [27], is an assembly of protein-protein interactions and regulations from multiple databases and computational predictions (including BioGrid [28]). Therefore, it is larger in size, but it potentially contains many spurious interactions, observed only in a certain context or predicted by computational biology methods. In order to compare interactions between networks, each Wikipedia protein page title was matched to a standard HUGO gene symbol.

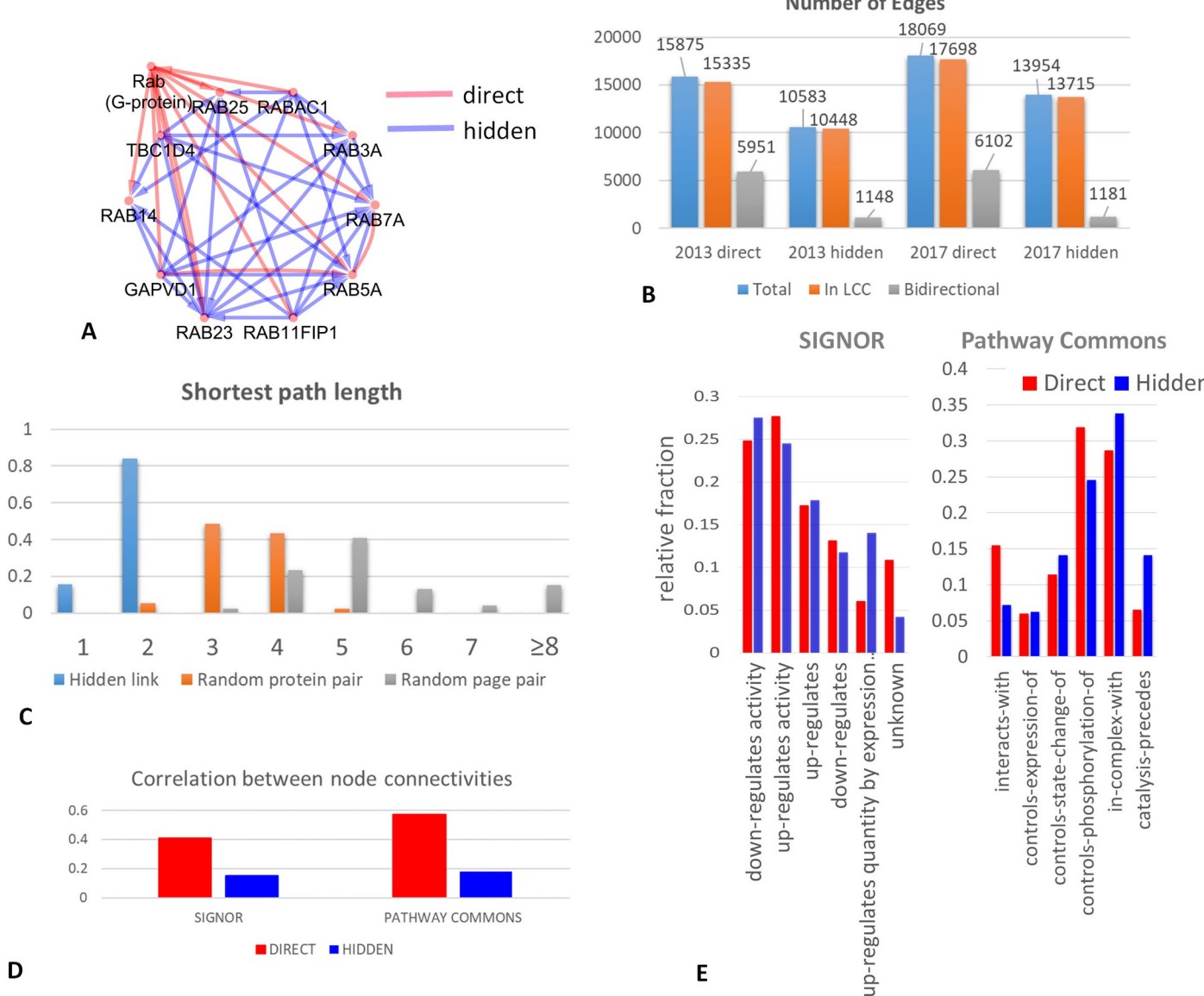

**Fig 1. Characterizing the networks of direct and hidden connections between proteins in the global Wikipedia network.** A) Example of a distinction between direct and hidden interactions. The page "Rab (G-protein)" links to a set of pages for the proteins from the family. Hidden interactions (in blue) connects the whole family into an almost complete clique graph, making them a tight community. B) Number of edges in the networks of direct and hidden protein connections. Number of bidirectional links is separately shown. C) Quantifying the shortest path length in the global Wikipedia network between proteins connected by a hidden link, a random protein page pair and a random Wikipedia page pair. D) Correlation between node degrees in the networks of direct (or hidden) protein connections extracted from Wikipedia and two pathway databases (SIGNOR and Pathway Commons). E) Relative fraction of link types found in the networks extracted from Wikipedia (direct and hidden) and in two pathway databases (SIGNOR and Pathway Commons).

The overlap between the links extracted from Wikipedia and the pathway databases was not very strong but highly significant. Thus, for SIGNOR we found 1714 proteins in common with Wikipedia. These proteins were connected in SIGNOR by 4026 interactions from which we found 861 direct and 170 hidden links matched in Wikipedia. Of note, on average 10% of direct protein links are also contained in the hidden protein link network, i.e. they are 'direct+hidden' links. However, in this comparison, there was a strong bias and among 170 matched hidden links there were 105 'direct+hidden' links. For Pathway Commons, we found

4768 proteins in common with Wikipedia protein page list. These proteins were connected in Pathway Commons by 269665 interactions from which we found 8212 direct and 2563 hidden links (of which 795 'direct+hidden') matched in Wikipedia. Taking as a null hypothesis that any two proteins from the Wikipedia network could be connected, this gives statistical significance for a Fisher's exact test with p-values of the order of $10^{-300}$, in all comparisons. The significance of the overlap is, however, not completely surprising, given that a number of direct interactions between proteins were automatically imported from BioGrid pathway database (for example, around 60% of direct links between protein pages are also found in BioGrid). The intersection size between the set of direct links and existing pathway databases is, however, small in absolute numbers, but it is comparable to the expected intersection between independently built pathways databases in terms of the number of common links. The intersection between hidden links and pathways databases is expected to be smaller, because hidden links are not supposed to describe physical interactions but rather reflect the indirect functional proximity between proteins.

As a conclusion, a number of direct and hidden connections between proteins from Wikipedia are not found in the existing databases, which reflects relative independence and non-redundancy of two sources.

We verified subsequently if the node degree distribution for the nodes matched between Wikipedia network and pathway databases is similar. The comparison showed a significant correlation between the matched node degrees (see Fig 1D, which was much higher for the network of direct links (Pearson R = 0.4 and Pearson R = 0.58 for SIGNOR and Pathway Commons correspondingly) compared to the network of hidden links (Pearson R = 0.18 and Pearson R = 0.19 correspondingy). This correlation was determined, to a large extent, by the existence of common hubs in two types of networks. For example, BRCA1 was the top connected protein in the network of direct links form Wikipedia version 2017, and it ranks 35 in the top connected proteins in the network of Pathway Commons.

We further checked which interaction types are more present in those links which were matched between Wikipedia protein network and a pathway database. In order to do this, for each interaction type $t$, we first computed the fraction of matched interactions $f_t = I_t^W / I_t^{PD}$, where $I_t^{PD}$ is the total number of links in a pathway database $PD$ of type $t$ and $I_t^W$ is the number of these interactions matched in Wikipedia protein network. When computing $I_t^{PD}$, we limit the network only to those proteins common between Wikipedia and a pathway database. In order to compare direct and hidden connections, we used the relative fraction value $f_t^{rel} = f_t / \sum_t f_t$, which is shown in Fig 1E. From this comparison it emerges that some interaction types have higher chance to be found in the Wikipedia network (e.g., "down-" or "up-regulates activity" interaction type in SIGNOR). We also detected a difference between direct and hidden interactions with respect to which interaction type they match more frequently. For example, for the interaction type "catalysis-precedes" of Pathway Commons there is almost three-fold increase in the relative frequency of match with hidden interactions, while for the "interacts-with" type the relative match frequency is much higher for direct interactions. Also, it seems that the hidden interactions between proteins in the Wikipedia network reveal more frequently co-participation of proteins in a complex, compared to direct interactions.

## Community structure of the Wikipedia network of hidden connections between proteins

Simple visual inspection of the 2D force-directed layouts of networks of direct and hidden connections shows existence of relatively small scale compact communities in the network of hidden connections (Fig 2A). We compared the two networks, using three network topology

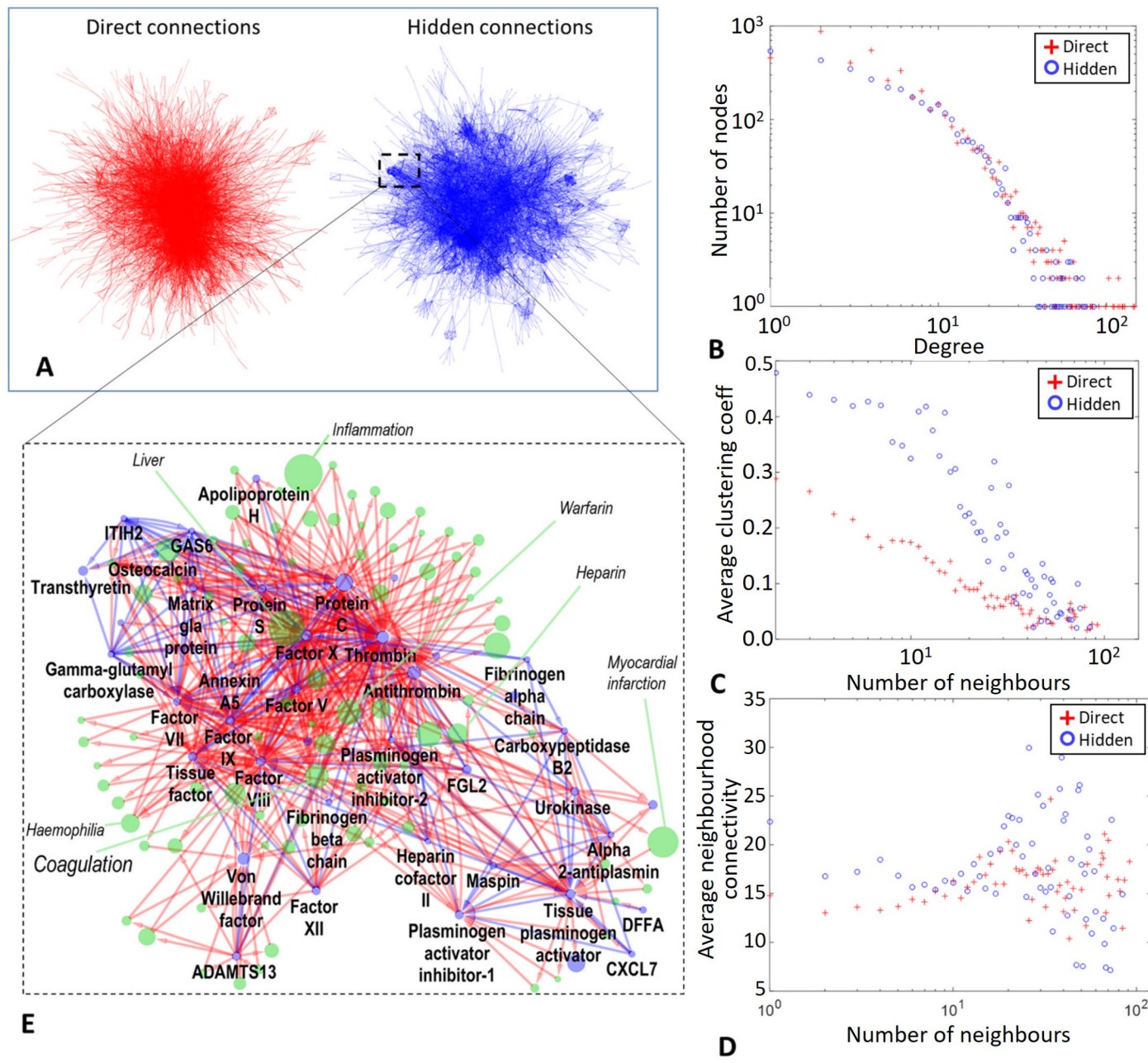

**Fig 2. Network of hidden interactions is characterized by relatively well defined communities as compared to the network of direct interactions.** A) Force-directed layouts of the networks of direct and hidden connections. B-D) Comparison of two networks in terms of node degree distribution, average clustering coefficient, average neighborhood connectivity distribution. E) One of the communities in the network of hidden interactions is shown together with direct links to the Wikipedia pages connecting the protein pages (the augmented or friendship network, composed of protein pages, shown in blue, and "associated" pages, shown in green).

measures, namely node degree distribution, average clustering coefficient distribution and average neighbourhood connectivity distribution (Fig 2B, 2C and 2D). For all three measures, the networks of direct and hidden interactions resulted to be similar for the nodes with large (more than 20) number of neighbours. At the same time, nodes having smaller number of neighbours (less than 20), are characterized by larger local connectivity in the case of the network of hidden protein connections. This is particularly pronounced for the average

clustering coefficient which equals 0.19 and 0.35 respectively for the direct and hidden connection networks, for the nodes having 10 neighbours (Fig 2C). This analysis allowed us to conclude that the hidden protein connection network is characterized by the presence of communities of variable sizes. We thus hypothesized that these communities could be matched to the biological functions implicitly defined in Wikipedia through the community-based effort.

## Annotated hidden protein connection community map

Following our conclusion about the presence of small communities in the hidden network, we clustered the network of hidden protein connections using Markov Cluster Algorithm [29]. 274 and 289 communities were identified in the network of hidden protein connections computed from the English Wikipedia graph from 2013 and 2017 correspondingly (only those communities having size at least 4 pages were kept in this phase). The maximum community size was 148 and 187 correspondingly for 2013 and 2017 Wikipedia versions. Despite this size of the largest community, overall the others resulted to be smaller with average community size 10 (median 8) and 12 (median 9) in 2013 and 2017 correspondingly. For comparison, we also applied MCL algorithm with the same default parameters to the network of direct connections (S2 Fig). In agreement with the results described above, the community size distribution was different for the networks of direct and hidden connections (S2 Fig). It reminded the power law distribution in the case of direct connections, while in the case of clustering the network of hidden connections, the tail of small size communities was much shorter than it would be expected from the power law.

Using HUGO symbols matched to the Wikipedia protein page titles, we performed function enrichment analysis for all communities using ToppGene [24]. The detailed results of this analysis are available online at [22]. We found that most of the communities had clear enrichment in one of the biological functions or in a biological pathway. Thus, the geometric mean q-value of the most significant enrichment in a Gene Ontology-related term was $10^{-19}$ (for communities with at least 10 proteins), and in a Pathway term it was $10^{-16}$. The exceptionally large community with 187 nodes (2017 version) had enrichment in Gene Ontology terms "cytokine activity" (q-value = $10^{-30}$), "leukocyte proliferation" (q-value = $10^{-55}$), "adaptive immune response" (q-value = $10^{-47}$), pathway terms "Cytokine-cytokine receptor interaction" (q-value = $10^{-50}$), "Hematopoietic cell lineage" (q-value = $10^{-39}$) and other multiple immune system-related terms. It was also strikingly enriched in the MSigDB HALLMARK [30] gene sets: e.g., "Genes up-regulated during transplant rejection" (q-value = $10^{-52}$), "Genes defining inflammatory response" (q-value = $10^{-28}$).

Alternatively to the use of the enrichment analysis, the biological function defined by a community could be identified by looking at the direct connections through neigbouring Wikipedia pages. For each link inside the community we extracted titles from the global Wikipedia graph along the shortest oriented paths of length 2 connecting the connected pair of proteins. This defined an augmented or friendship network, with Wikipedia pages corresponding not only to protein pages but also to the Wikipedia titles through which the shortest paths had gone through. An example of such an augmented network is shown in Fig 2E. We ranked the set of nodes by their node degree in the augmented network, and used the most connected page title for labeling the community. For example, the augmented network shown in Fig 2E was labeled in this way as "Coagulation". Also, among the most strongly locally connected nodes there were such titles as "Haemophilia", "Warfarin", "Heparin", "Liver" and others. The three most connected proteins in the augmented network shown in Fig 2E were "Protein C", "Thrombin", "Factor X".

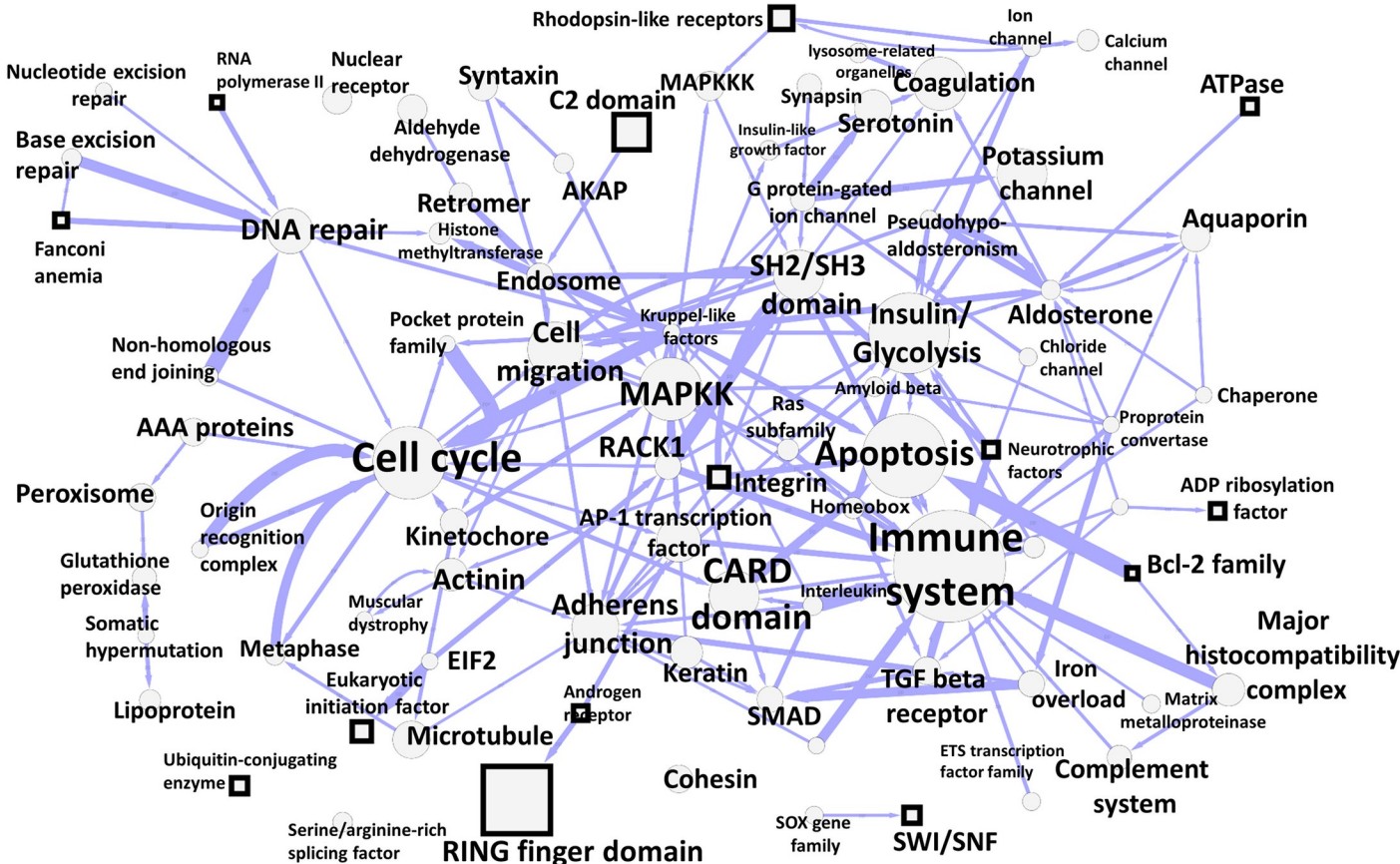

**Fig 3. Map of connections between communities of hidden protein connections through the global Wikipedia network.** The size of the nodes reflect the number of proteins in the community. Square nodes signify those communities identified in the 2017 Wikipedia network which do not have a match in the 2013 Wikipedia network. The width of the line reflects the number of oriented links between the communities. The node is labeled by the title of the Wikipedia page associated with the community by the connector links, having the largest node degree. The interactive version of this map is available from http://navicell.curie.fr/pages/maps_wikipediacommunity.html.

Following this strategy, we annotated each community by the Wikipedia title, having the largest node degree in the augmented network. In some cases, we manually changed this title, selecting among the 10 most connected titles, which would have a better match for the enriched biological function. Afterwards, we counted the number of hidden links between the nodes in each community from the initial network of hidden protein connections. In this way we constructed an abstracted graph of communities and oriented links between them, which we visualized in Cytoscape [31], using force-directed layout, Fig 3. This map shows the repertoire of biological functions described in English Wikipedia by groups of pages forming relatively compact subnetworks in the global graph of hyperlinks. As one can see, the central place in this map is taken by "Immune system", "Apoptosis", "Cell cycle", "Insulin/Glycolysis", "Mitogen-activated protein kinase kinase", "Cell migration" and other communities which correspond to the well studied biological functions. There exist relatively large protein page communities collecting proteins characterized by a presence of a particular domain such as "CARD domain", "RING finger domain", "SH2/SH3 domain", "C2 domain". Interestingly, the map is characterized by a meaningfull hierarchy of functions. For example, 4 communities annotated by the names of the major DNA repair pathways ("Non-homologous end joining",

"Nucleotide excision repair", "Base excision repair", "Fanconi anemia") point to a relatively large community annotated as "DNA repair".

We provide the hidden protein connection community map in interactive form, using NaviCell Google Maps-based platform for annotated network visualization [32, 33]. The online map of hidden protein interactions in Wikipedia is available from http://navicell.curie. fr/pages/maps_wikipediacommunity.html. The map can be queried for a protein name or a part of the Wikipedia page title. All community node annotations are hyperlinked to the corresponding Wikipedia pages. Therefore, the interactive map serves as a convenient portal to the set of Wikipedia pages related to proteins and associated pages. Moreover, the map can be used for molecular data visualization, using NaviCell data analysis toolbox and binding to major programming languages (R, Python, Java) [33].

## Evolution of Wikipedia protein network between 2013 and 2017

We compared the changes in the direct and hidden protein connections, between two versions of English Wikipedia (2013 vs 2017). We found that 96% of direct connections did not change in four years, while only 71% hidden connections remained unchanged in the same period of time (Fig 4A). This indicates that the Wikipedia network of protein connections evolves more slowly through the curation of pages devoted to proteins compared to more dynamic modifications of the information in the associated pages from the network neighbourhood (for example, pages describing molecular mechanisms of diseases or pages devoted to the systematic description of protein families). Of note, part of the changes in the direct protein links were caused by the process of translating the automated annotations produced by Protein Box Bot into human-curated article texts. For example, this explains the disappearance of a part of direct links in 2017 compared to 2013.

From the reduced Google matrix analysis we know the relative PageRanks of proteins which were not exactly the same between two Wikipedia versions, despite good overall correlation (Fig 4B). Thus, we found that a significant number of proteins strongly improved their PageRanks in 2017 (S1 Fig). For example, MGMT gene changed its PageRank from 1856 to 174 (more than ten-fold) and FANCB gene changed its PageRank from 3240 to 351 (almost ten-fold). Such drastic changes in PageRanks might indicate recent interest in studying these genes which led to intense curation of the associated pages.

We verified if such proteins with drastically improved PageRank were enriched in a particular biological function. To answer this question, 181 proteins whose PageRank improved more than two-fold were tested for the enrichment in reference gene sets using ToppGene. In the top of the list of the enriched categories we found such Gene Ontologies as "fibroblast growth factor receptor binding" ($p = 10^{-6}$), "damaged DNA binding" ($p = 10^{-5}$), "response to radiation" ($p = 10^{-10}$), "aging" ($p = 10^{-9}$), "DNA repair complex" ($p = 10^{-7}$), "transcription factor complex" ($p = 10^{-6}$). Among MSigDB signatures, the top enriched was "Genes involved in DNA repair, compiled manually by the authors" (19 genes, $p = 10^{-10}$). Overall, it shows significant recent editing efforts in the part of Wikipedia related to DNA repair, which led to higher hidden connectivity between pages in this area. At the same time, 46 genes loosing their PageRank position more than 2-fold did not show any strong enrichment in Gene Ontologies or other reference gene sets (e.g., none has passed the corrected p-value threshold 0.001).

We matched the communities obtained in 2013 and 2017 versions of Wikipedia by computing the Jaccard index for the overlap between the set of the genes composing them. We defined a match, if the Jaccard index was reciprocally maximal between two community sets aka it is done for defining orthologous genes in evolutionary bioinformatics [34]. Overall, 189 communities could be matched in this way with a minimum threshold for the intersection in 3

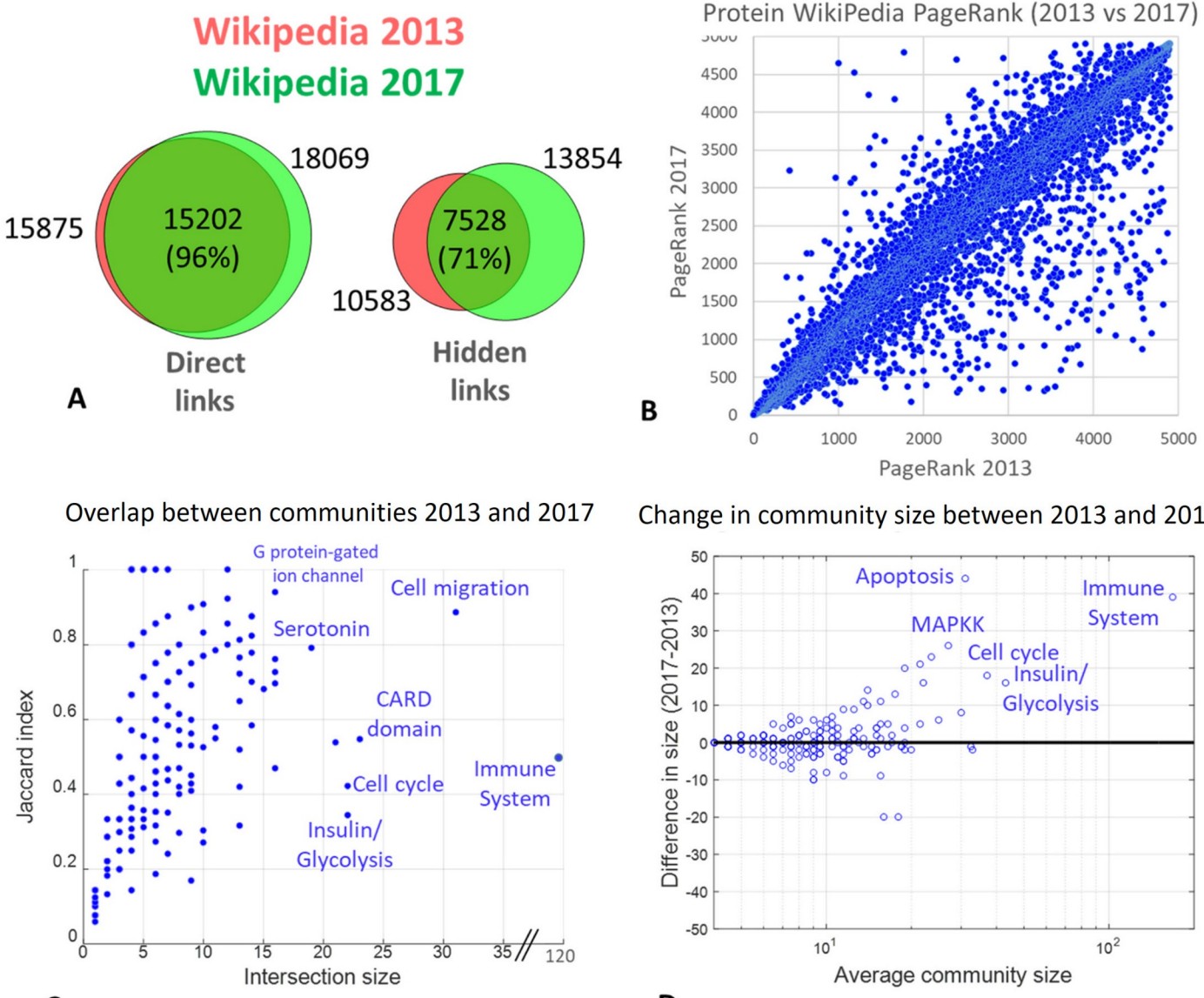

**Fig 4. Evolution of the networks of protein connections within the global Wikipedia network between 2013 and 2017.** A) Overlap between links for the network of direct and hidden protein connections in two versions of Wikipedia. B) Changes in the PageRanks in the reduced Google matrix for protein pages, compared between 2013 and 2017. C) Overlap between matched communities of hidden protein connections extracted from two versions of Wikipedia. D) Change in matched hidden interaction community size between two versions of Wikipedia.

proteins. We observed a consistent increase between the matched community sizes between 2013 and 2017 versions, starting from the community size in 10 protein pages, Fig 4D.

The abstracted map of hidden protein connection communities shows emergence of some communities in the 2017 version of English Wikipedia which can not be matched in 2013 version. Examples are "RING finger domain", "SWI/SNF", "ATPase", "Bcl-2 family", "Integrin", "Fanconi anemia" communities (Fig 3). Hypothetically, this also indicates an active curation efforts happening between 2013 and 2017 in the Wikipedia pages related to these functions or the pages directly connected to them.

## Analysing Wikipedia contributor set defining direct and indirect protein connections

In order to verify our statement, put in the article title, that the emerging structure of indirect protein connections through the rest of the Wikipedia network is the result of the action of the wisdom of the crowd or the collective intelligence principle, we looked in more details at the set of Wikipedia contributors involved in updating the corresponding pages.

For this purpose, we defined two sets of pages: those of proteins involved in the identified communities ('protein pages'), and the set of pages from the augmented friendship network for each community ('associated pages'). For each page in these sets, we listed all contributing Wikipedia users having status 'confirmed' or 'extended confirmed', i.e. those users whose identity can be established through their nicknames. This filtering also excluded numerous automated Wikipedia bots and resulted in the set of 12380 users (which we denote as 'all contributors') made at least a single contribution to a protein or associated page. However, most of them made non-specific contributions (formatting, style, etc.), therefore, we prioritized users working specifically on these page sets. In order to do this, we generated a set of random Wikipedia page titles of comparable size (10000 pages), called 'random pages', and listed all 'confirmed' or 'extended confirmed' users contributing to them. After this, we defined the restricted list of 'expert' contributors accordingly to two criteria: Activity) a user should contribute to at least 10 pages from protein or associated page set; Specificity) the number of contributions to the random page set should be at least ten times smaller than to the protein and associated page set. This resulted in the set of 889 'experts'.

In order to confirm their expertise, we looked at the Wikipedia pages (when they existed) of all these users, and counted all Wikipedia categories in which they were listed. We noticed that the most frequent categories included those related to medicine, biology and science, and made manual selection of 30 of them. Fig 5A shows this list and the fraction of expert users labeled by the selected categories. For comparison, we looked at the categories of 1000 'non-specific contributors', i.e. those who contributed to random page set at least ten times more frequently than to the protein and associated page set. Fig 5A clearly shows that the set of 'expert' users is strongly enriched in medicine- and biology-related categories. Among those of them having user pages in Wikipedia, 46% were labeled by at least one category listed in Fig 5A. We also verified that the proportion of WikiGnomes (i.e., those users practicing systematic minor grammatic and stylistic changes in Wikipedia pages) was identical in both 'expert' and 'non-specific contributor' user sets (an expert contributor can be a WikiGnome at the same time).

We further looked at the user workload and found that the efforts of both 'all contributors' and 'expert' groups, quantified as the number of pages to which they contributed, were distributed accordingly to power law (Fig 5B), as expected from previous reports [21]. We identified two distinguished supercontributor or 'hub' experts, 'Boghog' and 'Dcirovic', who contributed to 72% and 49% of protein and associated pages accordingly. Remarkably, manual inspection showed that many of their contributions were related to the actual scientific content of the pages and not to their styles. In the five top supercontributor experts the real identity of two was disclosed. These are Andrew Su (computational biology professor from The Scripps Research Institute) and Mikael Höggström (physician at NU Hospital Group), contributed to 15% and 13% of protein and associated pages correspondingly.

From the other hand, the activity of hub users remained less important when compared to the total number of 'contributor-page' pairs (individual contributions). For the protein pages, the effort of the first two supercontributing users ('Boghog', 'Dcirovic') explained 30% of total-lindividual contributions, which was still quite noticable. However, their total contribution

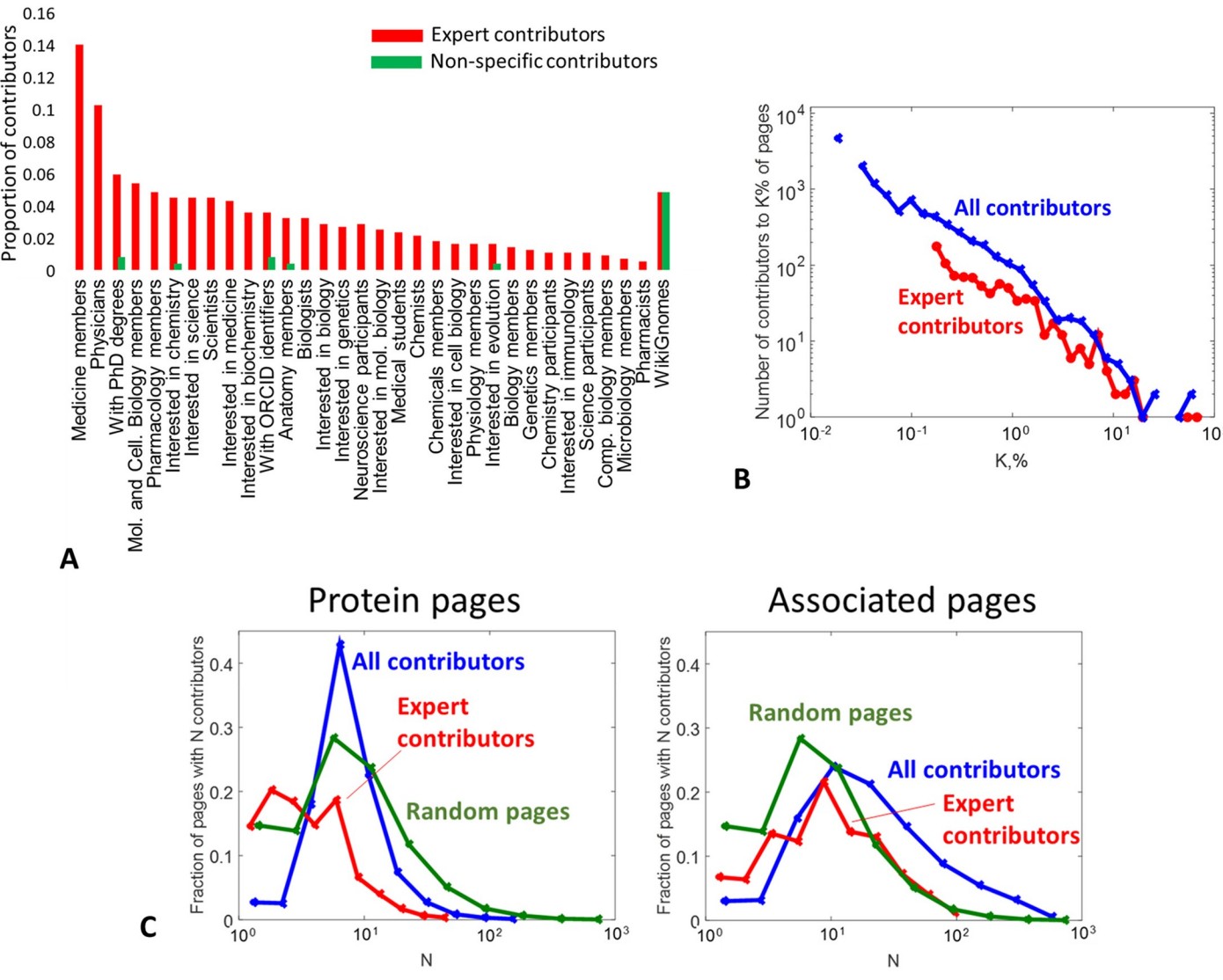

**Fig 5. Analysis of WikiPedia contributor set defining direct and indirect protein connections.** A) Proportion of contributors listed in a biology- or medicine-related WikiPedia category. List of 889 'experts' (active users specifically working on protein and associated page set) is compared to 1000 'non-specific contributors' (those who contributed more to a random set of pages). B) Distribution of efforts in terms of percentage of pages from the protein and associated page set, for all contributors and the expert contributors. C) Distribution of the number of contributors per page shown separately for protein page set and for associated page set, distinguishing all and expert contributors. For reference, the distribution of the number of contributors for a set of 10000 random WikiPedia pages is shown in both plots.

was only 8% to the total number of individual contributions to associated pages, which we use as the estimation of the contribution to the formation of indirect links between proteins within the hidden protein communities. The rest of 92% individual contributions was a result of a collective effort, distributed over the whole set of experts, even if with unequal efforts following approximately the power law. In other words, it was not possible to distinguish a small group of experts dominating the definition of the structure of indirect links between proteins, which we interpret as a feature of the collective intelligence.

We finally looked at the general level of protein and associated page curation. We estimated that a random page in Wikipedia is written by 7 'confirmed' and 'extended confirmed' contributors as a median. The protein pages were characterized by the same 7 median number of

contributors, but only 3 of them were 'experts'. By contrast, the associated pages had 16 contributors as a median, with 9 'experts' (see Fig 5C). This indicates relatively high level of interest of both general massive and specialized expert Wikipedia contributors to the set of associated pages, rather than to protein pages. This can partly explain more rapid evolution, reported by us above in the text (Fig 4A), of the indirect links between the proteins in Wikipedia compared to direct links.

## Discussion

We studied the network of protein-protein interactions embedded into the graph of hyperlinks between Wikipedia pages. We focused on comparing direct hyperlinks between protein pages (most of which were automatically imported from existing molecular interaction databases) and hidden links through the rest of the Wikipedia graph. The hidden links were identified by using the reduced Google Matrix (REGOMAX) approach.

The most striking conclusion from this analysis is the existence of pronounced small-scale (containing 10-20 proteins) clusters (communities) in the network of hidden protein connections. The absolute majority of these clusters have rather clear biological meaning which was quantified by the functional enrichment analysis. This is in contrast with the previous conclusions about approximate power law node degree distribution of protein pages in the global Wikipedia network [9]. Existence of such clusters (communities) points out to emergence, due to collective intelligence of Wikipedia contributors, of relatively well defined groups of proteins sharing the common biological function (such as cell cycle), structural feature (such as SH2/SH3 domain) or other common topics, as it was confirmed in this study by the systematic functional enrichment analysis. These clusters are generally not present and can not be deduced from the network of direct interactions.

Interestingly, one can deduce the biological function of the community by looking at the titles of the pages most tightly connected in the the augmented network of pages linking the community proteins (protein friendship network). Using this labeling, we created an abstracted interactive online map of connections between the protein communities, which can serve a portal to the Gene Wiki Wikipedia project.

We characterized the evolution of the network of hidden protein connections and its community structure between two snapshots of Wikipedia in 2013 and 2017 years. We showed that the nature of hidden protein connections is much more dynamic compared to the direct links. A clear trend has been noticed on the faster relative increase of the number of hidden connections such that they combine more proteins in one largest connected component. This can be partly explained by the relatively strong interest of the active Wikipedia contrubutors having biomedical background to the pages connecting proteins. We estimated the size of this expert community in approximately 1000 contributors. Interestingly, we showed that there are more proteins that drastically (by few folds) improved their PageRanks in 2017 compared to those who drastically lost their PageRanks inside the global Wikipedia network. We found that the Wikipedia topics being improved in node degree were related to DNA repair and damage. Most of the hidden connection network communities between 2013 and 2017 can be matched in terms of maximally reciprocal Jaccard index quantifying their intersection. We showed that the matched communities have larger size on average in 2017 compared to the 2013 network.

Altogether, these observations indicate increasing integration of the Gene Wiki project into the global Wikipedia context, a trend which will certainly persist in the future. It remains an interesting question what can be a practical use of of the protein function definition derived from the Wikipedia structure. Another interesting question is how to use the insights obtained from analysing the topology of hidden protein connections, in order to guide further evolution

of the Gene Wiki project. For example, it would be interesting to identify missing biological functions or topics which do not yet form tight clusters in the Wikipedia network.

## Supporting information

**S1 Fig. Parameter analysis of the Markov Clustering (MCL) algorithm.** Clustering of the graph of hidden connections between proteins through WikiPedia network was repeated 18 times for various combinations of inflation and expansion parameters, including the default one (inflation = 2, expansion = 2). For the application of each MCL run, a list of clusters larger than 4 proteins was ranked by size. On the left, graph shows intersections between the clusters obtained using the multiple runs of the MCL algorithm. Each node in the graph represents a cluster, labeled in the form 'cl_(Inflation)_(Expansion)_(Cluster Number)' (smaller cluster numbers corresponds to the largest clusters, the numbering starts from zero). The node size is proportional to the size of the cluster. The edges in the graph represent intersections between the clusters characterized by Jaccard similarity index larger than 0.4. The width of the edge is proportional to the value of Jaccard similarity index. On the right, zooming on two groups of clusters is shown, corresponding to the communities "Immune system" and "Cell cycle" from Fig 3. Each combination of Inflation/Expansion parameters in each group is characterized by the proportion of cell cycle genes (624 genes from REACTOME Cell cycle pathway) and immune-related genes (1170 genes from GO 'Regulation of immune process') correspondingly, and the cluster size.
(TIF)

**S2 Fig. Distribution of community sizes in the direct and hidden protein link networks.** The MCL custering algorithm was applied with the same parameters to the largest connected components of both networks. Community size distribution is shown in the left panel. Cumulative number of the communities larger than a certain size is shown in the right panel.
(TIF)

**S1 File. Table containing computed PageRanks of Wikipedia protein pages within the reduced network, definitions of hidden protein connection communities.**
(XLSX)

**S2 File. Cytoscape sessions containing networks of direct and hidden connections between proteins, in 2013 and 2017.**
(ZIP)

## Author Contributions

**Conceptualization:** Andrei Zinovyev, Dima L. Shepelyansky.

**Data curation:** Andrei Zinovyev, Urszula Czerwinska, Laura Cantini, Dima L. Shepelyansky.

**Formal analysis:** Andrei Zinovyev, Laura Cantini, Klaus M. Frahm.

**Funding acquisition:** Andrei Zinovyev, Emmanuel Barillot, Dima L. Shepelyansky.

**Investigation:** Andrei Zinovyev, Urszula Czerwinska, Laura Cantini.

**Methodology:** Andrei Zinovyev, Urszula Czerwinska, Laura Cantini, Klaus M. Frahm, Dima L. Shepelyansky.

**Resources:** Andrei Zinovyev, Emmanuel Barillot.

**Software:** Andrei Zinovyev, Klaus M. Frahm, Dima L. Shepelyansky.

**Supervision:** Andrei Zinovyev, Dima L. Shepelyansky.

**Validation:** Andrei Zinovyev, Laura Cantini.

**Visualization:** Andrei Zinovyev, Urszula Czerwinska.

**Writing – original draft:** Andrei Zinovyev, Dima L. Shepelyansky.

**Writing – review & editing:** Andrei Zinovyev, Urszula Czerwinska, Laura Cantini, Emmanuel Barillot, Klaus M. Frahm, Dima L. Shepelyansky.

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
