## [Decision Letter · Decision Letter 0]

27 Sep 2019

Dear Dr Zinovyev,

Thank you very much for submitting your manuscript, 'Collective intelligence defines biological functions in Wikipedia as communities in the hidden protein connection network', to PLOS Computational Biology. As with all papers submitted to the journal, yours was fully evaluated by the PLOS Computational Biology editorial team, and in this case, by independent peer reviewers. The reviewers appreciated the attention to an important topic but identified some aspects of the manuscript that should be improved.

We would therefore like to ask you to modify the manuscript according to the review recommendations before we can consider your manuscript for acceptance. Your revisions should address the specific points made by each reviewer and we encourage you to respond to particular issues Please note while forming your response, if your article is accepted, you may have the opportunity to make the peer review history publicly available. The record will include editor decision letters (with reviews) and your responses to reviewer comments. If eligible, we will contact you to opt in or out.raised.

- Supporting Information uploaded as separate files, titled 'Dataset', 'Figure', 'Table', 'Text', 'Protocol', 'Audio', or 'Video'.

We hope to receive your revised manuscript within the next 30 days. If you anticipate any delay in its return, we ask that you let us know the expected resubmission date by email at ploscompbiol@plos.org.

Sincerely,

Andrey Rzhetsky

Associate Editor

PLOS Computational Biology

Feilim Mac Gabhann

Editor-in-Chief

PLOS Computational Biology

[LINK]

Reviewer's Responses to Questions

**Comments to the Authors:**

Reviewer #1: In this article, Zinovyev et al. investigated the English Wikipedia corpus to construct networks of proteins. They especially compared the network built upon Wikipedia to the established knowledge database on protein interactions (SIGNOR and Pathway Commons) and found significant though relative low proportion of overlap. They asserted that “hidden connections” (indirect connections) between proteins tend to form compact communities in small scale. Also, they studied the evolution of the network from 2013 to 2017.

It is my understanding that this study is a good visualization of collective intelligence on biology. There are a few technical details I want to discuss with the authors:

When defining the hidden communities by graph clustering, the authors use Markov clustering with default hyper-parameters. In many cases, the results of clustering is very sensitive to the setting of hyper-parameters, did the authors try other combinations of setting and did the results look quite different? The current results consist of more than 250 communities, which could be reduced for more concise visualization.

In Line 227 and Fig. 4A, the authors asserted that the hidden links changed more the the direct links through evolution, this is understandable because the polyline connections are more fragile. What may me more interesting is what changed in the direct connections (4%). Did these reflect alteration of knowledge or findings? I notice that some of the connections in 2013 does not exist in 2017, does this have any particular meaning?

Reviewer #2: In the paper "Collective intelligence defines ..."

authors show that the hidden links (derived from connections of greater than the first order) connecting proteins in Wikipedia map well the direct protein network (which is imported from databases of protein-protein interactions).

There are several issues the authors have to address before the paper can be accepted for publication:

a) Throughout the paper the authors refer to node degree as `node connectivity`, if I am not mistaken. I would suggest replacing node connectivity by node degree wherever applicable, since node degree is a more popular way to refer to the number of neighbors (as well as to avoid confusion with graph connectivity [the minimum number of elements (nodes or edges) that need to be removed to separate the remaining nodes into isolated subgraphs].

b) Page 3, lines 118-124.

1)

Do we understand why the overlap between Wikipedia and databases is so low?

2)

Direct and hidden links are not mutually exclusive, is it correct? Then how to understand 861 direct and 170 hidden links matched to SIGNOR? 861 direct and 170 hidden which are not direct? Or some of the 861 direct might be hidden and vice versa?

3) It seems that metrics like precision and recall would be more adequate than Fisher test against the null hypothesis that any two protein can be connected.

c) Page 6, line 267. Please provide the details of the web-crawler and possibly the code public.

After the points above are addressed the paper may be suitable for publication.

Reviewer #3: The manuscript describes properties of the directed network of proteins contained in Wikipedia, focusing on two snapshot versions of 2013 and 2017. The manuscript is generally well written although some sentences appear rushed (see line 267 for an example).

A major drawback of the manuscript is the lack of an articulated problem or question to be solved: the line of argument is that proteins and genes on Wikipedia can be represented as a directed weighted network, and that network methods can be applied to study the network. What is missing is a goal, however tentative, that would give a direction of both the narrative and the computations. Such a goal would also help biologists comprehend why they should be interested in this description.

Choices of design, methods, and measures often receive little to no justification. For example, no clear justification is given to why only two snapshot versions of Wikipedia are considered, and why the focus is on versions of 2013 and 2017 (and whether/how those choices can impact the results). Choices are not unreasonable (e.g., PageRank, the community detection algorithm, Jaccard index for pairwise cluster intersection) but there are still many alternatives, and within those alternatives different parameters could be selected, all of which would affect the descriptive statistics. For example, community size is known to depend on the resolution of the algorithm used to discover those communities, so one needs to evaluate communities across multiple sets of parameters to be able to convincingly show that community size is stable across parameters and is therefore a feature of the network, not the algorithm.

The main conclusions appear suggestive but not fully supported. For example, the authors concluded:

"The most striking conclusion from this analysis is the existence of pronounced small-scale (10-20 proteins on average) clusters (communities) in the network of hidden protein connections. The absolute majority of these clusters have rather clear biological meaning which was quantified by the functional enrichment analysis. This is in contrast with the previous conclusions about the power law-like distribution of connectivity of protein pages in the global Wikipedia network.”

Unfortunately, no frequency distribution plot appears to show the number of clusters with a particular size. Only average values appear to be provided. The authors report clear outliers—clusters (in the networks of hidden protein connections) with size of 150 or more proteins in the Wikipedia versions of 2013 and 2017 (see line 173). The size of those largest communities in 2013 and 2017 does not rule out the possibility of a heavy-tailed distribution. Further, if the power-low argument is considered essential, a more informed approach would be to directly evaluate whether the distribution is power law-like or not rather than to infer it from the mean number of communities. Further, in the presence of outliers, the median (instead of the mean) would likely give a more realistic estimate of the average community size.

An association between community structure and biological function is strongly postulated but is only tentatively supported by suggestive examples. See for example lines 331–333:

"Existence of such clusters (communities) points out to emergence, due to collective intelligence of Wikipedia editors, of relatively well defined groups of proteins sharing the common biological function (such as cell cycle), structural feature (such as SH2/SH3 domain) or other common topics.

A major question is whether the clusters/communities have emerged due to collective intelligence or due to a few very active contributors, which are often considered to be a defining feature of Wikipedia. Essentially, the paper has not examined the distribution of editors producing the connections between pages, and has not ruled out the hypothesis that the clustering of proteins is due to a few very active editors that formed direct and/or indirect connections between many pages. Thus, a key property of the gene Wikipedia network noted in the title—collective intelligence—is assumed to be true but is not directly examined.

**Have all data underlying the figures and results presented in the manuscript been provided?**

Reviewer #1: None

Reviewer #2: Yes

Reviewer #3: Yes

PLOS authors have the option to publish the peer review history of their article (what does this mean?). If published, this will include your full peer review and any attached files.

Reviewer #1: No

Reviewer #2: No

Reviewer #3: No

---

## [Editor Report · Decision Letter 1]

13 Jan 2020

Dear Dr Zinovyev,

We are pleased to inform you that your manuscript 'Collective intelligence defines biological functions in Wikipedia as communities in the hidden protein connection network' has been provisionally accepted for publication in PLOS Computational Biology.

In the meantime, please log into Editorial Manager at https://www.editorialmanager.com/pcompbiol/, click the "Update My Information" link at the top of the page, and update your user information to ensure an efficient production and billing process.

One of the goals of PLOS is to make science accessible to educators and the public. PLOS staff issue occasional press releases and make early versions of PLOS Computational Biology articles available to science writers and journalists. PLOS staff also collaborate with Communication and Public Information Offices and would be happy to work with the relevant people at your institution or funding agency. If your institution or funding agency is interested in promoting your findings, please ask them to coordinate their releases with PLOS (contact ploscompbiol@plos.org).

Thank you again for supporting Open Access publishing. We look forward to publishing your paper in PLOS Computational Biology.

Sincerely,

Andrey Rzhetsky

Associate Editor

PLOS Computational Biology

Feilim Mac Gabhann

Editor-in-Chief

PLOS Computational Biology

---

## [Editor Report · Acceptance letter]

12 Feb 2020

PCOMPBIOL-D-19-01094R1 

Collective intelligence defines biological functions in Wikipedia as communities in the hidden protein connection network

Dear Dr Zinovyev,

I am pleased to inform you that your manuscript has been formally accepted for publication in PLOS Computational Biology. Your manuscript is now with our production department and you will be notified of the publication date in due course.

With kind regards,

Sarah Hammond
